# A Framework for Vision-Language Warm-up Tasks in Multimodal Dialogue Models

**Jaewook Lee,  Seongsik Park,  Seong-heum Park,  Hongjin Kim,  Harksoo Kim**
Konkuk University
{benecia428, a163912, tjdgma95, jin3430, nlpdrkim}@konkuk.ac.kr

## Abstract

Most research on multimodal open-domain dialogue agents has focused on pretraining and multi-task learning using additional rich datasets beyond a given target dataset. However, methods for exploiting these additional datasets can be quite limited in real-world settings, creating a need for more efficient methods for constructing agents based solely on the target dataset. To address these issues, we present a new learning strategy called vision-language warm-up tasks for multimodal dialogue models (VLAW-MDM). This strategy does not require the use of large pretraining or multi-task datasets but rather relies solely on learning from target data. Moreover, our proposed approach automatically generates captions for images and incorporates them into the model's input to improve the contextualization of visual information. Using this novel approach, we empirically demonstrate that our learning strategy is effective for limited data and relatively small models. The result show that our method achieved comparable and in some cases superior performance compared to existing state-of-the-art models on various evaluation metrics. The code is available at https://github.com/BeneciaLee/VLAW-MDM

## 1 Introduction

Developing artificial intelligence (AI) that can converse naturally with humans is a primary goal of AI research. In particular, open-domain conversational agents that are not restricted to a specific domain have attracted considerable attention. Many studies have adopted a pretraining approach using large datasets to improve the performance of these open-domain conversational agents(Adiwardana et al., 2020; Roller et al., 2021).

Recent works have focused on multimodal open-domain conversational agents that consider visual information in addition to textual information for dialog generation. This approach utilizes visual information to help understand the context of a

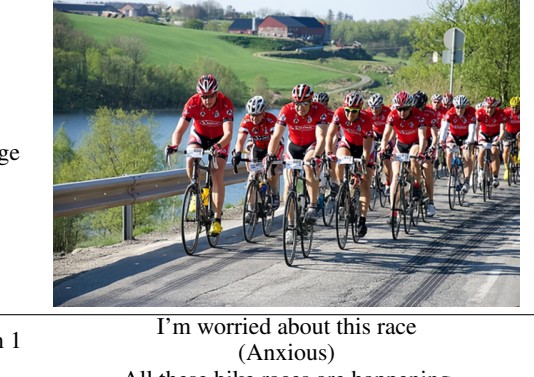

| | |
|---|---|
| Image | |
| Turn 1 | I'm worried about this race (Anxious) |
| Turn 2 | All these bike races are happening near my street. (Dull) |
| Turn 3 | I'd be worried about accidentally hitting one of them (Anxious) |

Table 1: IMAGE-CHAT dataset

conversation, which is more consistent with how humans communicate. This approach has been shown to be effective in generating conversations that users find more engaging(Hu et al., 2014). To build such multimodal open-domain conversational agents, most previous studies have considered multi-task learning or have utilized pretrained models using large-scale data beyond the target data(Shuster et al., 2020b, 2021). This allows models trained with only text information to accept visual information. However, the requirement of collecting additional datasets is restrictive, and pretraining with additional datasets is inefficient in terms of time and resources. In this study, we propose a method to align text and images using only the target data to address these issues. We experimentally evaluated the effectiveness of the proposed method with limited data or smaller models.

The proposed method utilizes only target data to align images and texts. By automatically generating captions for the images and adding them as input to the model, our approach enables a text-based pre-

trained model to process image information more effectively. This differs notably from existing multimodal open-domain models that only receive image and utterance information. We propose vision-language warm-up tasks for multimodal dialogue models (VLAW-MDM) to effectively integrate information from images, captions, and context data. To construct the framework, we incorporated four warm-up tasks based on existing multimodal pre-training models(Chen et al., 2020; Yu et al., 2021; Wu et al., 2022). These tasks include generation captioning (GCP), image swapping (ISP), masked region modeling (MRM), and masked language modeling (MLM). They can be applied using only target data without any additional data required. These warm-up tasks enable the model to learn the associations between images and utterances.

The construction of this framework was inspired by the generative framework used by Ling et al. (2022), which we reformulate into a multimodal architecture suitable for the purposes of this research. We apply our warm-up framework to the popular sequence-to-sequence models BART and Blender-Bot. For BlenderBot, we used a smaller model (400M) than that (2.7B) adopted in the multi-modal blenderBot (MMB)(Shuster et al., 2021). This allowed us to explore the performance of our proposed framework on smaller versions of the model and to validate the framework against a relatively large model with additional training data.

We used the Image-Chat dataset(Shuster et al., 2020a) to evaluate the effectiveness of our proposed framework. The data were structured as shown in Table 1. Image-Chat comprises conversations organized into a series of turns with utterances based on the speakers' styles. We used Image-Chat for warm-up tasks first, and then performed fine-tuning to evaluate the effectiveness of our proposed method in a constrained learning environment.

The main contributions of this study are summarized as follows:

- We propose a framework for vision-language warm-up tasks in multimodal dialogue models, called VLAW-MDM, and describe the process of warming up the model using only data from the target task. We experimentally evaluated the performance of this framework as described above.

- We introduce four different warm-up tasks (MLM, ISP, MRM, and GCD) and experimen-

tally evaluated how they each affected the performance of the model. Our results show that the best performance was achieved when all four warm-up tasks were utilized together.

- We analyzed how automatically generating and utilizing captions affected the performance of the model. Our results showed that our proposed framework incorporating caption information was effective for training a multimodal open-domain dialogue model.

- We also evaluated the warm-up tasks in the absence of caption information. The results show that the proposed method is effective even in environments where captions are not available or are difficult to create.

## 2 Related Work

**Vision and Language Tasks.** The integration of language and vision is a topic of active research that traditionally includes tasks such as image captioning and visual question answering (VQA)(Devlin et al., 2015; Xu et al., 2015; Fang et al., 2015; Donahue et al., 2015; Antol et al., 2015; Ray et al., 2016; Malinowski and Fritz, 2014). Image captioning tasks focus on generating appropriate descriptions of a given image. Major datasets include COCO Captions (Chen et al., 2015) and Flickr30k (Young et al., 2014). These datasets of images covering various topics provide an ideal benchmark for assessing model's ability to understand complex content in an image and express it in natural language. Sequence-to-sequence structures are the most common method to process these datasets (Vinyals et al., 2015; Xu et al., 2015; Anderson et al., 2018).

The VQA task (Antol et al., 2015) requires image recognition and factual verification of text content. It evaluates the ability of a model to generate accurate answers to questions related to a given image. As a natural extension of this work, a method has been proposed to generate questions that can capture a user's attention based on a given image. However, these methods involve the limitation that the conversation usually ends in a single turn. To address this, visual dialog(Das et al., 2017) extends this with a continuous question-answer scenario.

However, this approach does not provide a direct way to evaluate whether users experience conversations as interesting and engaging. To address this, multimodal multi-turn dialog datasets

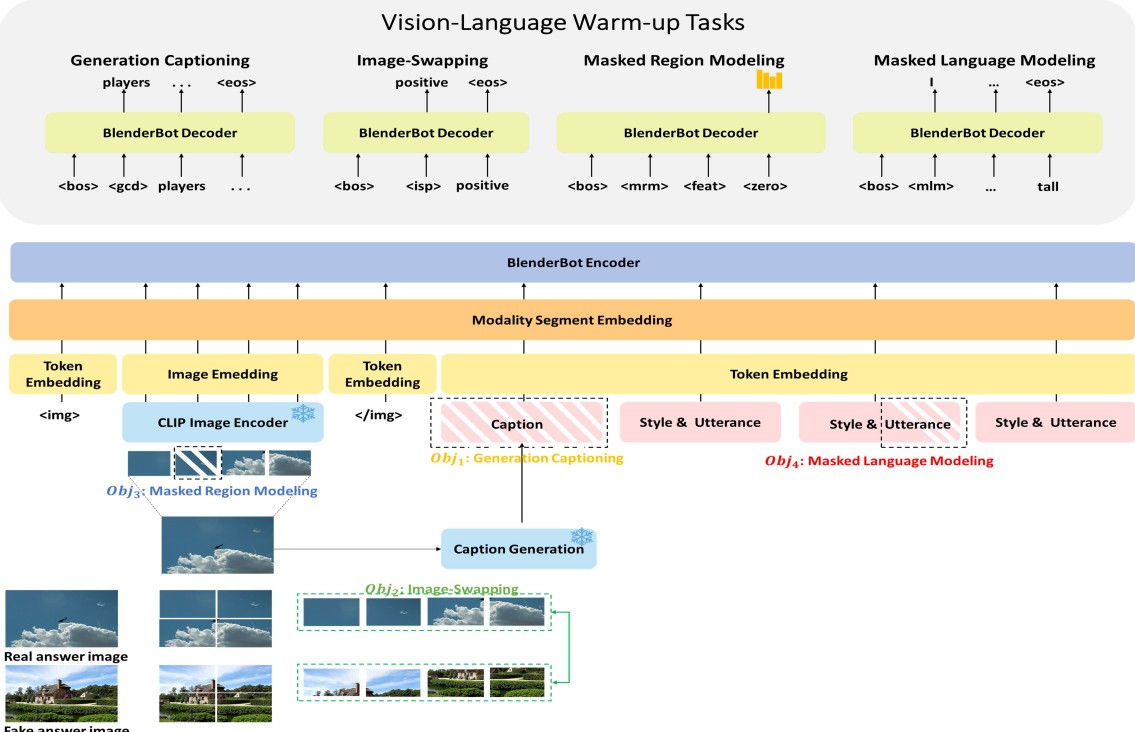

Figure 1: Overview of the Proposed Framework for Vision-Language Warm-up Tasks in Multimodal Dialogue Models.

(Mostafazadeh et al., 2017; Shuster et al., 2020a) have been proposed. In particular, Image-Chat (Shuster et al., 2020a) supports multi-turn dialog based on a single image. Each utterance contains style information to allow a model to learn different styles of conversation. Hence, Image-Chat is an ideal dataset for training multimodal open-domain agents and enables more engaging dialog generation.

**Multimodal Representation Learning.** Utilizing the weights of existing models trained on a single modality and fusing them together is a common strategy for multimodal learning(Kiela et al., 2019; Le and Hoi, 2020; Chen et al., 2020; Yu et al., 2021; Zhang et al., 2021). Many studies have adopted this approach to reuse models pretrained on a single modality for multimodal representation learning.

To process images and text together, Shuster et al. (2020b) uses multi-task training, in which a model learns by bundling multiple tasks that are related to a given target task. For this purpose, Shuster et al. (2020b) includes 12 subtask sets, which allows it to perform multiple tasks on large datasets.

In some cases, multimodal representations have been learned from multimodal datasets(Li et al.,

2020a; Chen et al., 2020). However, multimodal pretraining approach has generally not been performed with data in the target domain. One potential solution is to employ domain-adaptive pretraining by using data related to the domain of the target data(Shuster et al., 2021). This method enables a model to adapt more effectively to a specific domain. However, domain-adaptive pretraining also utilizes data related to the target domain and does not provide a pretraining methodology for specific target data. As a solution, Ling et al. (2022) proposed a task-specific vision-language pretraining framework for multimodal aspect-based sentiment analysis, which realizes target data-specific pretraining on multiple tasks instead of a single task.

## 3 Methodology

In the present work, we adopt BlenderBot as a backbone model. The overall architecture is illustrated in Figure 1. In this section, we describe the operation of the entire framework. First, we describe how we process the images used as input to the model. In particular, we discuss our approach to extract features from an image and generate captions. Then, we describe how the encoder is extended to handle multimodal inputs and how the decoder

generates utterances from the information received from the encoder. Finally, we describe the warm-up tasks that comprise the framework.

## 3.1 Image Encoder

Our proposed method uses pretrained models to extract visual features from images. Previous studies(Shuster et al., 2021; Ling et al., 2022) have mainly used Faster R-CNN(Anderson et al., 2018). Recently, patch-based models have shown better performance in image encoding(Shen et al., 2021; Wu et al., 2022). Based on these findings, we adopt a patch-based method based on CLIP in the proposed approach(Radford et al., 2021). In image encoding, a single image is divided into nine patches used as input to the model to obtain visual features. We denote the visual features as $R = \{r_1, \ldots, r_9\}$, where $r_i \in \mathbb{R}^{512}$ is the visual feature of the $i$-th patch. The obtained visual features do not have the same number of dimensions as the textual representation, so an additional linear transformation layer is used to put the visual features along with the textual representation as input to the multimodal encoder. This linear transformation layer projects the visual features of each patch to a $d$-dimensional vector denoted as $V \in \mathbb{R}^{d \times 9}$.

## 3.2 Caption Generation

Image captions are textual descriptions of objects, situations, and other information provided by an image. We used captions as a bridge between images and text. Because a caption is a textual representation of the information in an image, we assume that aligning the image with the utterance text is beneficial. However, because there are no separate captions for images in the existing dataset, we use an image captioning model (Li et al., 2023) to generate captions. The generated captions provide a description of the image and are used as input to the multimodal encoder along with the image and utterance.

## 3.3 Multi-Modal Architecture

**Encoder.** The encoder receives different kinds of modality information. To separate the modality information, we add a segment embedding that separates the image from the text. We also add special tokens such as $\langle img \rangle$, $\langle /img \rangle$ before and after the extracted image features following Xing et al. (2021). As shown in Figure 1, the images are entered in the order they appear first in the modality information. The image feature is followed by the caption created earlier. There is no special token for the caption; rather a $\langle sep \rangle$ token is simply added to the end of the caption. The caption is followed by the style and utterance. An additional special token such as $\langle sty \rangle$ is appended at the end of styles to distinguish them from utterances. Styles are followed by utterances, and the difference between the warm-up task and fine-tuning phases becomes relevant here. In the warm-up task phase, the styles and corresponding utterances are input to the encoder together. However, in the fine-tuning phase, the style is not followed by an utterance because the model needs to predict an utterance for the style. Therefore, in the warm-up task, styles and utterances are combined and followed by an $\langle eos \rangle$ token. However, during the fine-tuning stage, the $\langle eos \rangle$ token is appended immediately after the $\langle sty \rangle$ special token representing the style.

**Decoder.** As shown in Figure 1, all warm-up tasks are processed through a single decoder. To distinguish between warm-up tasks, we add a special token at the beginning of the decoder's input, following a prior work (Yan et al., 2021; Ling et al., 2022). The input of the decoder starts with $\langle bos \rangle$, followed by $\langle gcp \rangle$, $\langle isp \rangle$, $\langle mrm \rangle$, and $\langle mlm \rangle$, depending on the warm-up task. The input format is followed by the label values according to the warm-up task.

## 3.4 Warm-up Tasks

In this study, we introduce VLAW-MDM to efficiently integrate multimodal information. This is a warm-up task that strengthens the connections between images and text before the fine-tuning phase to improve the model's ability to handle complex multimodal information more effectively. During this warm-up task, the model is trained to understand and strengthen the relationship between images and text by utilizing data on the target task. This improves the model's ability to process multimodal input, which in turn improves its performance by utilizing only data on the target task without any additional data for pretraining. These enhanced connections between images and text play an important role in improving performance on the target task during the fine-tuning phase, allowing for more effective utilization of multimodal data at no additional cost.

**Generation Captioning (GCP).** The GCP task replaces all captions with masking tokens and restores them to the original captions. In the GCP

task, the model interprets the context of the image based on other multimodal information such as image or utterance data without any caption information and generates caption accordingly. This helps the decoder not only analyze and understand information from each modality independently but also acquire the ability to comprehensively understand and appropriately integrate information from other modalities such as images and utterances.

The target sequence for the GCP task is $Y = [\langle gcd \rangle, c_1, \dots, c_N, \langle eos \rangle]$, where $c$ represents caption tokens and $N$ is the number of caption tokens. Traditional training methods such as maximum likelihood estimation (MLE) have a problem in that they mainly generate high-frequency responses that exist in the dataset. Therefore, to control these high-frequency responses, we adopt unlikelihood training (Roller et al., 2021; Li et al., 2020b; Welleck et al.). The formula for MLE is as follows:

$$\mathcal{L}_{MLE} = -\sum_{t=1}^{|Y|} log p_\theta(y_t | \widetilde{X}, y_{<t}), \quad (1)$$

where $y_t \in Y$ and $y_t$ is a caption token or special token for $Y$. Let $y_{<t}$ be the tokens before the $t$-th utterance of $y_t$. The $\widetilde{X}$ means that the caption input to the encoder is masked, where the rest of the multimodal information except for the caption token is in its normal form. The formula for the unlikelihood loss function is as follows:

$$\mathcal{L}_{UL} = -\sum_{t=1}^{|Y|} \sum_{y_c \in C_t} log(1 - p_\theta(y_c | \widetilde{X}, y_{<t})), \quad (2)$$

The negative candidates $C_t$ are the set of tokens that we do not want to generate at each time step. This is controlled by assigning a penalty if the token generated by the model belongs to $C_t$. Likelihood is used to increase the probability of the next fired token, $y_t$, while unlikelihood is used to decrease the probability of $y_c$. The final loss value for GCP is as follows:

$$\mathcal{L}_{GCP} = \mathcal{L}_{MLE} + \alpha \mathcal{L}_{UL} \quad (3)$$

$\alpha$ is the weighting representing how much to reflect $\mathcal{L}_{UL}$.

**Image-Swapping(ISP).** The ISP task serves to train the model's ability to determine whether an image is the original or an altered image. The main process of an ISP task is as follows. An image is replaced with another image in the batch with a certain probability. Images are then fed into the encoder along with caption and dialog. The encoder processes this multimodal information and passes the results to the decoder. Based on this information, the decoder determines whether the image is an original or an altered image. The results are expressed as "positive" or "negative." "positive" means that the decoder recognizes the original image, whereas "negative" means that the image has been altered.

The target sequence for the ISP action is $Y = [\langle bos \rangle, \langle isp \rangle, S, \langle eos \rangle]$, where $S$ indicates "positive" or "negative." The loss function for the ISP task is as follows:

$$\mathcal{L}_{ISP} = -\mathbb{E}_{X \sim D, I = X \cup \widetilde{X}} \sum_{t=1}^{|Y|} \log p_\theta(y_t | I, y_{<t}), \quad (4)$$

$y_t \in Y$, which refers to a text token or special token in $Y$. Let $y_{<t}$ be the tokens before the $t$-th utterance of $y_t$.

In the loss function, we consider two cases: $X$ and $\widetilde{X}$. $X$ is an instance from a data distribution $D$ that retains the original image, caption, and utterance. In contrast, $\widetilde{X}$ represents a case where only the image changes while the caption and utterance remain the same. $I$ denotes the combined input information for these cases, encompassing both $X$ and $\widetilde{X}$.

**Masked Region Modeling(MRM).** We adopted the MRM method used by Xing et al. (2021); Ling et al. (2022). We masked random positions in the patches. The masked image is passed to a multimodal encoder, and the decoder estimates the masked part(Ling et al., 2022). For the MRM task, the inputs to the decoder are $\langle feat \rangle$ and $\langle zero \rangle$. Here, the $\langle feat \rangle$ token is used for the unmasked normal image region and the $\langle zero \rangle$ token is used for the masked part, which feeds the value at position $\langle zero \rangle$ into the MLP layer. The MLP layer is trained to match the output representation to the original image representation. The final loss value of MRM is as follows:

$$L_{MRM} = -\mathbb{E}_{X \sim D} \Sigma_{r=1}^{R} D_{KL}(q_{(v_r)} \| p_{(v_r)}), \quad (5)$$

The representation predicted by the model is $p(v_r)$ and the actual input image representation is $q(v_r)$. The model was trained by minimizing the KL divergence, where $R$ is the number of masked regions.

**Masked Language Modeling(MLM).** For MLM, we masked the tokens in an utterance at

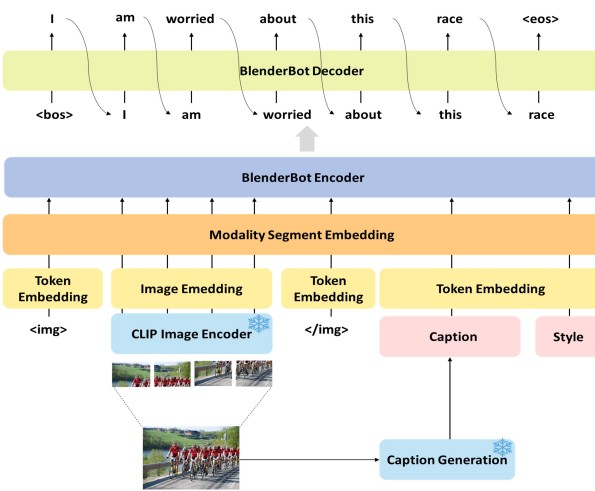

Figure 2: Example of turn-by-turn utterance generation for Image-Chat.

| | Train | Valid | Test |
|---|---|---|---|
| Number of Images | 186,782 | 5,000 | 9,997 |
| Number of Utterances | 355,862 | 15,000 | 29,991 |
| Style Types | 215 | 215 | 215 |
| Vocabulary Size | 46,371 | 9,561 | 13,550 |
| Token per Utterance | 12.3 | 12.4 | 12.4 |

Table 2: IMAGE-CHAT dataset statistics.

a certain rate. In this case, masking was not performed for the entire turn but rather only for a certain percentage in each turn(Devlin et al., 2018).

The target sequence for the MLM task is $Y = [\langle mlm \rangle, d_1^{s_1}, \ldots, d_N^{s_1}, \langle sty \rangle, d_1^{u_1}, \ldots, d_M^{u_1}, \langle eos \rangle]$. The sequence consists of the style of the utterance and the turns that represent the utterance. It begins with a start token $\langle bos \rangle$ and $\langle mlm \rangle$, followed by $N$ tokens $d^s$ indicating the style, and ending with the token $\langle sty \rangle$ token to indicate the end of the style. This is followed by $M$ tokens $d^u$ representing the utterances in that turn, with the turns separated by $\langle sep \rangle$ tokens. In this example, we indicate the end of the turn through the $\langle eos \rangle$ token directly, without a separate $\langle sep \rangle$ token, because it indicates that we sampled for Turn 1.

To calculate the MLM loss value, the loss value is calculated as in the GCP operation, and the final loss value is as follows:

$$\mathcal{L}_{MLM} = \mathcal{L}_{MLE} + \alpha \mathcal{L}_{UL} \quad (6)$$

**Full Warm-up Task Loss.** The final objective function is as follows:

$$\mathcal{L} = \lambda_1 \mathcal{L}_{GCP} + \lambda_2 \mathcal{L}_{ISP} + \lambda_3 \mathcal{L}_{MRM} + \lambda_4 \mathcal{L}_{MLM} \quad (7)$$

The $\lambda$ values given above are adjustable hyperparameter. For this experiment, all $\lambda$ values were fixed to 1.

## 4 Experiment

### 4.1 Experimental Setup

**Evaluation Metrics.** We used the F1, BLEU-4, and ROUGE-L metrics to evaluate the performance of the proposed model.

**Fine-tuning the Dataset.** We used the Image-Chat dataset to verify the effectiveness of the proposed framework. The data consisted of an image, style attributes of two speakers (A and B), and a dialogue between the two speakers. It also included a set of 215 possible style attributes from Shuster et al. (2019), which are categorized as positive, neutral, and negative. The style attributes are used to define the speakers' personalities in the conversation. The images in the dataset were selected from the YFCC100M dataset (Thomee et al., 2016). Some statistics on the data included in the Image-Chat dataset are as shown in Table 2.

The utterance generation for the first turn using Image-Chat is shown in Figure 2. The data input to the encoder in the first turn were the image, caption, and style of the first turn. The encoder processes these inputs and passes the information to the decoder. The decoder generates an utterance for the first turn based on the information from the encoder. In the second turn, the encoder receives an image along with a caption and a style from the first turn, and utterance of the first turn, and a style from the second turn. The decoder generates the second round of utterance from these inputs, and the third round proceeds in the same manner.

### 4.2 Main Results

**Impact of Each Warm-up Task.** Table 3 shows the contribution of each warm-up task to performance. We compared the performance of models trained with and without warm-up tasks.

To validate the scalability of the warm-up task, we applied it to two different sequence-to-sequence models, including BART and BlenderBot. First, in terms of BlenderBot's performance, an improvement may be observed in all measures except the BLEU-4 score at Turn 3 when the MLM task was introduced compared to no warm-up task. This was likely due to the word prediction ability learned through the MLM helping with utterance generation. Next, when we added the ISP task alongside MLM, we observed additional performance

| Model | Warm-up Task | Turn 1 | | | Turn 2 | | | Turn 3 | | | IC | | |
|---|---|---|---|---|---|---|---|---|---|---|---|---|---|
| | | F1 | B | R | F1 | B | R | F1 | B | R | F1 | B | R |
| BART | w/o warm-up task | 8.97 | 0.44 | 10.41 | 12.34 | 0.60 | 10.37 | 14.68 | 0.79 | 11.70 | 12.00 | 0.61 | 10.83 |
| | MLM | 11.16 | 0.61 | 9.90 | 12.92 | 0.65 | 9.96 | 14.39 | 0.75 | 11.12 | 12.82 | 0.67 | 10.33 |
| | MLM+ISP | 11.76 | 0.70 | 10.32 | 13.12 | 0.66 | 10.32 | 14.41 | 0.78 | 11.32 | 13.10 | 0.71 | 10.65 |
| | MLM+ISP+MRM | 11.77 | 0.67 | 10.08 | 13.26 | 0.65 | 10.17 | 14.61 | 0.77 | 11.42 | 13.21 | 0.70 | 10.56 |
| | MLM+ISP+MRM+GCD | 12.22 | 0.69 | 10.63 | 13.70 | 0.75 | 10.79 | 14.68 | 0.85 | 11.60 | 13.53 | 0.76 | 11.00 |
| BlenderBot | w/o warm-up task | 15.36 | 0.81 | 11.47 | 16.53 | 0.79 | 12.31 | 16.53 | 0.71 | 12.50 | 16.14 | 0.77 | 12.09 |
| | MLM | 15.72 | 0.90 | 11.69 | 16.75 | 0.86 | 12.43 | 16.88 | 0.70 | 12.70 | 16.45 | 0.82 | 12.27 |
| | MLM+ISP | 15.81 | 0.89 | 11.81 | 16.89 | 0.88 | 12.59 | 17.25 | 0.81 | 13.04 | 16.65 | 0.86 | 12.48 |
| | MLM+ISP+MRM | **15.91** | 0.91 | 11.90 | 17.02 | 0.87 | 12.70 | 17.24 | 0.78 | 13.09 | 16.72 | 0.85 | 12.56 |
| | MLM+ISP+MRM+GCD | 15.90 | **1.04** | **12.00** | **17.05** | **1.03** | **12.73** | **17.30** | **1.00** | **13.17** | **16.75** | **1.02** | **12.63** |

Table 3: Ablation study results presenting the performance of models across various warm-up tasks. Each model's performance was evaluated at three distinct interaction turns (Turn 1, Turn 2, Turn 3), each with their respective F1, BLEU-4(B), and ROUGE-L(R) measurements. The IC column represents the average of these metrics across all interaction turns.

| Caption | Warm-up Task | Turn 1 | | | Turn 2 | | | Turn 3 | | |
|---|---|---|---|---|---|---|---|---|---|---|
| | | F1 | B | R | F1 | B | R | F1 | B | R |
| w/o caption | w/o warm-up task | 14.85 | 0.91 | 11.20 | 16.16 | 0.89 | 12.12 | 16.42 | 0.84 | 12.48 |
| | MLM + ISP + MRM | 15.25 | 1.02 | 11.61 | 16.38 | 0.97 | 12.27 | 16.76 | 0.97 | 12.75 |
| w/ caption | w/o warm-up task | 15.36 | 0.81 | 11.47 | 16.53 | 0.79 | 12.31 | 16.53 | 0.71 | 12.50 |
| | MLM + ISP + MRM + GCD | **15.90** | **1.04** | **12.00** | **17.05** | **1.03** | **12.73** | **17.30** | **1.00** | **13.17** |

Table 4: Performance Evaluation based on caption information. Each model's performance is evaluated at three distinct interaction turns (Turn 1, Turn 2, Turn 3), each with their respective F1, BLEU-4 (B), and ROUGE-L (R) measurements.

gains on all scales except for the BLEU-4 score on Turn 1. ISP determines the appropriateness of a given image based on its caption and dialog. This allows the model to learn to align the image with the dialog, which is likely the reason for the performance improvement with the addition of ISP. Third, when we added the MRM task to MLM and ISP, the results showed a further increase in performance. MRM helps the model understand the image features given in the form of patches. This is important for multimodal open-domain agents that perform conversations based on images and seems to have helped with utterance generation. Finally, the highest performance was achieved when the GCD task was added to MLM, ISP, and MRM. The GCD task generates captions based on a given conversation and image. Through the GCD task, the model learns the caption and its relationship to a given conversation and image. This process allows the model to quickly learn the relationship between dialog, image, and caption. In particular, the information provided by the image during utterance generation appears to help the language model recognize and generate the correct utterance.

These experimental results show that each warm-up task contributed to improving the performance

of the model, and the best performance was achieved when all the warm-up tasks were combined. This demonstrates the effectiveness of the framework proposed in this study.

Table 3 shows that applying a warm-up task improved the performance of both BART and Blender-Bot. Compared to BlenderBot, BART is a smaller model. These results demonstrate that the proposed method is effective even for small models, as initially assumed.

**Impact of Caption.** Table 4 compares the performance of the models tested with and without caption. In Table 4, row 1 shows the performance of the backbone model without caption and row 3 gives the performance of the backbone model with caption. Comparing the two, it may be observed that simply providing caption helped the backbone model generate utterances. This was likely the case because the caption effectively acted as a bridge between the image and the utterance. Of note, the improvement on Turn 3 was smaller than that on Turns 1 and 2. When generating an utterance for Turn 3, the input was an utterance from Turns 1 and 2. As with the caption, the utterances in Turns 1 and 2 provide information about the image in the form of text, which made it easier for the language

model to understand the image and generate an utterance. This was most likely the reason that the two performances were similar.

The results with and without the warm-up task when no captions were provided are shown in rows 1 and 2. We excluded the GCD from the warm-up task because they were not captioned. The results show that the warm-up task without captions helped improve performance. These results show that the proposed method is effective even in environments where captions are not available or cannot be automatically generated.

The performance on the warm-up task when captions were provided is shown in rows 3 and 4. Because captions were provided, we ran all the warm-up tasks, including GCD. The results showed that applying all warm-up tasks significantly improved performance. In particular, the performance for Turn 3 on row 4 shows that we achieved a sufficiently high performance improvement compared with the other methods. This suggests that warming up the model with GCD is more effective than naively entering textual information (captions and turn-by-turn utterances) for images.

### 4.3 Experimental results compared to baseline

To evaluate the performance of the proposed method, we conducted a comparative analysis of Image-Chat with various existing models used in the experiments. The results of the comparison are listed in Table 5. Most of the compared models use additional datasets other than the data of the target task to perform pretraining to align text and image information, or apply various multi-task techniques. This differs from our model, which utilizes only data on the target task. See Appendix A for a description of models compared.

Table 5 shows that BlenderBot with our framework exhibited performance comparable to that of state-of-the-art methods. In particular, the highest performance was achieved in terms of F1 score. The proposed model is smaller than the BlenderBot model used by MMB. Nevertheless, our model outperformed the F1 score of MMB, which was previously the highest-scoring model, and also performed better in terms of BLEU-4 score. Consequently, these results show that our framework is able to incorporate image information into a model pretrained using only existing text data. They also demonstrate that our framework can be effectively applied to small models and can further improve

| Model | IC | | |
|---|---|---|---|
| | F1 | B | R |
| DialoGPT (Zhang et al., 2020) | 6.2 | 0.1 | 5.2 |
| Dodeca (Shuster et al., 2020) | 12.9 | **2.1** | **24.6** |
| 2AMMC (Ju et al., 2019) | 9.3 | 0.1 | 11.0 |
| BlenderBot (Roller et al., 2020) | 9.2 | 0.1 | 12.3 |
| MultiModal BlenderBot (Shuster et al., 2021) | 13.1 | 0.4 | 18.0 |
| VLAW-MDM | **16.8** | 1.0 | 12.6 |

Table 5: Comparison results with existing models for Image-Chat.

utterance generation by utilizing additional caption information.

## 5 Conclusion

We have proposed VLAW-MDM as a methodology for training multimodal open-domain agents using only target data to obviate the need for large amounts of data for pretraining or multiple tasks. The experimental results have shown that even with limited data, a model pretrained from a single modality can effectively process multimodal information. Furthermore, our proposed approach outperformed existing models in terms of F1 score on the Image-Chat dataset and outperformed MMB in terms of F1 and BLEU-4 scores despite its smaller size. In future work, we plan to explore extensions to this framework.

## Limitations

While the methodology presented in this study provides meaningful results, it also involves a number of limitations. To demonstrate the performance of the proposed framework, we only used Image-Chat, which is characterized by combining image and style information to perform multi-turn conversations. Therefore, differences in style and dialog format may have affected the findings of this study.

Because our model utilizes image captions as an additional input, it is highly dependent on the accuracy of the generated captions. Captions play an important role in the learning and performance of the model because they serve as a textual representation of the image. However, errors in the caption generation process or captions that do not

accurately reflect the key content of an image can affect the model's ability to generate utterances.

Finally, we quantitatively evaluated the performance of the proposed model. However, although this quantitative evaluation is useful for measuring overall performance, it involves some limitations in capturing qualitative aspects such as the user experience. These qualitative factors such as user satisfaction, convenience, and understanding are useful to more accurately evaluate the actual performance of a learning model. They also play an important role in improving models based on user feedback.

In this study, we have focused on a quantitative evaluation to clearly demonstrate the performance of our proposed methodology. However, we acknowledge that this does not comprehensively cover the qualitative factors. In future work, we plan to perform a qualitative evaluation based on user feedback to further evaluate the performance of the model and user satisfaction.

## Acknowledgements

This work was supported by Institute of Information & Communications Technology Planning & Evaluation(IITP) grant funded by the Korea government(MSIT) (RS-2023-00216011, Development of artificial complex intelligence for conceptually understanding and inferring like human). And this research was supported by the MSIT(Ministry of Science and ICT), Korea, under the ITRC(Information Technology Research Center) support program(IITP-2021-2016-0-00465) supervised by the IITP(Institute for Information & Communications Technology Planning & Evaluation). And this work was supported by Institute of Information & communications Technology Planning & Evaluation (IITP) under the metaverse support program to nurture the best talents (IITP-2023-RS-2023-00256615) grant funded by the Korea government(MSIT).

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

## A Compare with existing models

**DialoGPT(Zhang et al.)**: Additional social media data was used to perform conversational neural response generation from the GPT-2 model. This model can only take textual information as input.

**Dodeca(Shuster et al., 2020b)**: The multi-task learning approach was used to train multiple tasks at once, and for this purpose, the dodecaDialogue dataset was built. The dodecaDialogue dataset consists of 12 tasks, and the model was trained on these tasks. For image feature extraction, the ResNeXT-IG-3.5B model (Mahajan et al., 2018) was used.

**2AMMC(Ju et al., 2019)**: The model was constructed by combining ResNeXt-IG-3.B with Faster R-CNN image feature extraction. 2AMMC is utilized as a search model that references various transformers to blend ResNeXt-IG-3.5B and Faster R-CNN image features.

**BlenderBot(Roller et al., 2021)**: BlenderBot is a 2.7B-sized model with a sequence-to-sequence structure. It was also pretrained on a large dataset. It only takes a single modal representation, text, as input.

**Multi-Modal BlenderBot(Shuster et al., 2021)**: This is a multimodal extension of the BlenderBot model. MMB used BlenderBot's 2.7B model and domain-adaptive training to allow a model trained on a single modality to receive multimodal information.

## B Implementation Details.

BlenderBot was used as the backbone model. Unlike MBB, we did not use the 2.7B BlenderBot model, but rather adopted a smaller 400M BlenderBot model. The default hyperparameter values of the model were used without modification for comparison with MBB. The warm-up task was trained for 20 epochs, batch size was set to 16, and number of accumulation steps was set to 126. The model trained with the warm-up task is fine-tuned for the target task, Image-Chat. For fine-tuning, set epoch to 10, batch size to 32, and accumulation step to 8. For BART, the warm-up task runs for 10 epochs, the batch size is set to 32, and the accumulation step is set to 2. For fine-tuning, we set the epoch to 7 and the batch size to 64. We used AdamW(Loshchilov and Hutter, 2017) as the optimizer and additionally OneCycleLR(Smith and Topin, 2019). All implementations for this experiment were done via Pytorch.