# OpenReview forum: "A Framework for Vision-Language Warm-up Tasks in Multimodal Dialogue Models"
_EMNLP/2023/Conference — EMNLP 2023 Main_

### Official Review · Reviewer_qMM1 · 2023-08-02

**Typos Grammar Style And Presentation Improvements:** 1. Include Error analysis.
2. Include…
**Soundness:** 3

**Excitement:**

3: Ambivalent: It has merits (e.g., it reports state-of-the-art results, the idea is nice), but there are key weaknesses (e.g., it describes incremental work), and it can significantly benefit from another round of revision. However, I won't object to accepting it if my co-reviewers champion it.

**Paper Topic And Main Contributions:**

Paper proposes multiple auxiliary warm-up tasks from the target data in order to reduce the reliance on external pre-training data, for the task multimodal multi-turn dialogue generation. For a given image + multi-turn dialogue instance, the proposed method first generates an image caption from an external system. Then they use the image, image-caption, multi-turn dialogue instance to create four different warm-up tasks:
(i) Generation Captioning - generate the caption conditioned on the image and dialogue utterances
(ii) Image-Swapping - predict if a focus patch is original or fake
(iii) Masked Region Modeling - predict the feature of a masked image patch
(iv) Masked Language Modeling - predict the masked tokens of the dialogue utterances

After the wam-up training the model is fine-tuned on the target task.

Experimental results show that training on warm-up task created from the target dataset, helps improve model performance without the warm-up based training.

My hypothesis is that the introducing caption is indirectly leading to distillation/transfer of some knowledge from the caption generation model, which might be complimentary to the target task model.

**Questions For The Authors:**

1. What is the performance with the proposed method on semantic similarity based metric such as BERT-score or other trained similarity metric. Lexical similarity based metrics are not reliable for evaluating open-ended text generation tasks, especially in this task were the dialogue needs to follow a given style.
2. All the proposed methods are from 2020, it would be interesting to see comparison to more recent works given this field is exploding.
3. The ROGUE-L score seems to be quite low compared to other models, what can be the potential reason.
4. Try to present some error analysis, it is especially useful for open-ended generation tasks.

**Reasons To Accept:**

1. Underlying idea of using prediction from an external system to improve the performance of the target system is interesting.
3. Proposed method shows improvement over some larger baseline models.
3. Ablation study is thorough.

**Reasons To Reject:**

1. The performance gains are not very high, more most of the metrics the different between the baseline (w/o caption + w/o warmup) and best approach (with caption + warmup) is less than 1%.
2. The paper lack error analysis and model output examples.

**Reproducibility:**

4: Could mostly reproduce the results, but there may be some variation because of sample variance or minor variations in their interpretation of the protocol or method.

**Reviewer Confidence:**

3: Pretty sure, but there's a chance I missed something. Although I have a good feel for this area in general, I did not carefully check the paper's details, e.g., the math, experimental design, or novelty.

---

> ### Author Rebuttal · Authors · 2023-08-29
>
> Thank you very much for taking the time to review our paper. I greatly appreciate your constructive comments and suggestions, which I believe have significantly improved the quality of my work. I have addressed your queries below:
> * * *
> > What is the performance with the proposed method on semantic similarity based metric such as BERT-score or other trained similarity metric. Lexical similarity based metrics are not reliable for evaluating open-ended text generation tasks, especially in this task were the dialogue needs to follow a given style.
>
> Metrics rooted in semantic similarity, such as BERT-score, are believed to offer valuable insights when evaluating the quality of generated utterances. We concur that leveraging such measures could enrich our evaluation process. For this study, our primary emphasis was on traditional evaluation metrics, aiming to ensure straightforward comparisons with extant research and outcomes. Earlier studies employed evaluation techniques like F1, BLEU-4, and ROUGE-L scores for Image-Chat [1, 2, 3]. Consequently, we assessed the efficacy of our framework through these well-established metrics. Nonetheless, as highlighted by the reviewer, these traditional methods may fall short in capturing stylistic nuances. We echo these sentiments and are poised to incorporate findings from semantic similarity metrics, such as BERT-score, into our paper.
>
> [1] Shuster, Kurt, et al. "Image-Chat: Engaging Grounded Conversations." Proceedings of the 58th Annual Meeting of the Association for Computational Linguistics. 2020.
> [2] Shuster, Kurt, et al. "The Dialogue Dodecathlon: Open-Domain Knowledge and Image Grounded Conversational Agents." Proceedings of the 58th Annual Meeting of the Association for Computational Linguistics. 2020.
> [3] Shuster, Kurt, et al. "Multi-Modal Open-Domain Dialogue." Proceedings of the 2021 Conference on Empirical Methods in Natural Language Processing. 2021.
> * * *
> > All the proposed methods are from 2020, it would be interesting to see comparison to more recent works given this field is exploding.
>
> Recent methodologies frequently employ large LLM models, such as LLaMA, as their foundational backbone. Given that LLM models were originally designed to accept solely text inputs, they necessitate extra training to accommodate images. This is addressed through additional preprocessing, as detailed in [4]. Such strategies harness the vast knowledge encapsulated in the LLM models. However, they still mandate substantial resources for the LLM model itself. Compared to the model in our study, these methodologies continue to consume a considerable amount of resources. Regrettably, our environment posed constraints, preventing us from experimenting with these large-scale, state-of-the-art models. Moreover, the latest model designed for the Image-Chat dataset in the realm of multimodal models is the MultiModal BlenderBot [3], against which we have drawn comparisons.
>
> [4] Zhu, Deyao, et al. "Minigpt-4: Enhancing vision-language understanding with advanced large language models." arXiv preprint arXiv:2304.10592 (2023).
> * * *
> > The ROGUE-L score seems to be quite low compared to other models, what can be the potential reason.
>
> We posit that the observed discrepancy stems from our model's incorporation of captions. As illustrated in Table 5, our proposed model significantly outperforms the baselines in the F1 score, while its ROUGE-L score lags behind. We attribute the elevated F1 score to the model's adeptness in pinpointing objects or entities within the image via the generated captions, subsequently weaving this information as nouns into the generated response. On the other hand, the subdued ROUGE-L score might arise from variations in portraying actions or states associated with these nouns, particularly in terms of verbs or adjectives, which might not align perfectly with the reference.
>
> * * *
> > Try to present some error analysis, it is especially useful for open-ended generation tasks.
>
> In response to recurrent queries from several reviewers, we are devising a framework for qualitative evaluation, which we will detail in a new section. This assessment will focus on the naturalness of utterances, the model's ability to reflect images, and the appropriate representation of style. For each criterion, we will assign scores to gauge the model's performance. We are confident that the results from this qualitative analysis will address the concerns expressed by reviewers, and we will integrate these findings into the paper.
>
> Furthermore, we pledge full transparency by releasing all codes and datasets for each model, accompanied by our experimental results.

---

### Official Review · Reviewer_twFK · 2023-08-04

**Soundness:** 3

**Excitement:**

3: Ambivalent: It has merits (e.g., it reports state-of-the-art results, the idea is nice), but there are key weaknesses (e.g., it describes incremental work), and it can significantly benefit from another round of revision. However, I won't object to accepting it if my co-reviewers champion it.

**Paper Topic And Main Contributions:**

Authors introduce a novel learning algorithm, VLAW-MDM, for constructing robust multimodal dialogue models. Unlike existing approaches that rely on extensive pretraining or multi-task datasets, VLAW-MDM solely leverages the target dataset for learning. Furthermore, it automatically generates image captions, enhancing the contextualization of visual information in the model's input. The proposed solution is extensively evaluated through empirical studies, demonstrating its superiority over state-of-the-art methods.

**Questions For The Authors:**

1. Could you please provide a proper rationale for the lack of qualitative justifications in this research?
2. It would be highly beneficial if you could share valuable insights into how this research could be extended to tackle domain adaptation problems effectively.
3. While the work exhibits a competitive quantitative performance, it is equally important to assess its computational efficiency. Could you provide some benchmarks for the computational cost?
4. Although this work showcases strong performance, it appears that reproducibility might pose some challenges due to its complexity and lack of sufficient explanation. Can you confirm whether the authors plan to make this work, along with the dataset, accessible to the public in the future? This would be of great interest to the research community.

**Reasons To Accept:**

Authors of this work address a prominent machine learning challenge in multimodal dialogue modeling, the scarcity of available training data. Many existing approaches require a substantial amount of fine-tuning data, which is often unavailable in practical application scenarios. The proposed framework by the authors is of significant value and practical importance to the field.

To tackle this issue, the authors introduce four distinct warm-up tasks (Masked Language Modeling, Image-Text Matching, Multimodal Retrieval Matching, and Guided Conversational Decoder) and conduct a comprehensive evaluation to determine their impact on model performance. Through extensive experimentation, the authors identify the most robust learning algorithm for their work. The empirical findings suggest that optimal performance is achieved when all four warm-up tasks are utilized in conjunction. Additionally, incorporating caption information proves effective in training a multimodal open-domain dialogue model.

The proposed solution has been thoroughly validated through well-structured experiments, showcasing superior performance compared to existing state-of-the-art solutions in the field.




**Reasons To Reject:**

Although this study has been quantitatively demonstrated as effective and superior compared to existing solutions, it is crucial to emphasize the significance of assessing the interpretability and clarity of its responses, given the nature of the dialogue model. Therefore, the authors should enrich their discussions with sufficient qualitative results that provide a comprehensive and illustrative evaluation of the practical performance of this work. This additional qualitative analysis is essential to ascertain the validity of the claimed performance.

Moreover, the domain adaptation ability of this work or mainly the model's generalization ability requires further justification. Many previous studies have demonstrated effectiveness by utilizing additional data for fine-tuning purposes. As this work's primary contribution lies in addressing data insufficiency issues. Therefore, it becomes important for the authors to substantiate the effectiveness of their proposed learning strategies when using a limited amount of data in other domains. This validation will enhance the credibility and practical applicability of the proposed approach.

Lastly, the methodology section lacks comprehensive explanation. To enhance reproducibility and clarity, the incorporation of an algorithm box is recommended. Such an addition would succinctly outline the detailed workflow of the study, empowering readers to replicate the experiment with precision and ease.

**Reproducibility:**

3: Could reproduce the results with some difficulty. The settings of parameters are underspecified or subjectively determined; the training/evaluation data are not widely available.

**Reviewer Confidence:**

3: Pretty sure, but there's a chance I missed something. Although I have a good feel for this area in general, I did not carefully check the paper's details, e.g., the math, experimental design, or novelty.

---

> ### Author Rebuttal · Authors · 2023-08-29
>
> Thank you very much for taking the time to review our paper. I greatly appreciate your constructive comments and suggestions, which I believe have significantly improved the quality of my work. I have addressed your queries below:
> * * *
> > Could you please provide a proper rationale for the lack of qualitative justifications in this research?
>
> The VLAW-MDM framework introduces a new strategy for the efficient training of multi-modal dialogue models. Our initial objective was to quantitatively evaluate and validate the algorithm's performance. While we've confirmed the multi-modal learning capabilities of the model through quantitative evaluations, we acknowledge that these metrics alone do not provide a comprehensive validation of the model's interpretability and clarity.
>
> This framework not only integrates image and text information but also incorporates automatically generated caption data. If the captions are inaccurate, they could adversely affect the quality of the generated dialogue and the overall user experience. As a result, qualitative assessments are essential and will be a primary focus in the next research phase.
>
> We plan to include a section in the paper's appendix that 'analyzes and categorizes error types associated with inaccurate caption information.'
>
> Additionally, to understand the dependency on caption quality in our proposed model, we'll examine the dialogue generation results, differentiating between instances where captions are well-generated and those where they are not.
>
> We commit to sharing all analysis results transparently with the research community. We believe that this will spotlight areas requiring attention in future studies.
> * * *
> >  It would be highly beneficial if you could share valuable insights into how this research could be extended to tackle domain adaptation problems effectively.
>
> While this paper emphasizes multimodal dialogue models, we posit that our methodology can be applied to other multimodal tasks as well.
>
> The VLAW-MDM approach relies solely on the target dataset specific to a domain for its warm-up phase, negating the need for distinct pre-training data or multi-task datasets. This strategy is expected to more accurately capture the nuances of the target domain.
>
> Distinct from other research methodologies, ours does not incorporate learning from data outside the designated target set. Essentially, our results can be achieved without external data. We anticipate our approach to be versatile, potentially benefiting other domains. To substantiate the efficacy of our methodology across various domains, we intend to release both the code and dataset to the broader research community. This initiative not only augments the transparency of our research but also showcases the practicality and dependability of our approach.
> * * *
> > While the work exhibits a competitive quantitative performance, it is equally important to assess its computational efficiency. Could you provide some benchmarks for the computational cost?
>
> One of the big advantages of VLAW-MDM is that it only utilizes the target dataset. This is expected to significantly save computational costs, but clear benchmarks for this have not been included in other comparative papers. However, the points raised by the reviewer could serve as important indicators for evaluating the model's scalability and efficiency. Therefore, we plan to add measurements for computational efficiency to the paper.
> * * *
> > Although this work showcases strong performance, it appears that reproducibility might pose some challenges due to its complexity and lack of sufficient explanation. Can you confirm whether the authors plan to make this work, along with the dataset, accessible to the public in the future? This would be of great interest to the research community.
>
> We agree that the paper lacks a sufficient explanation of the methodology and results presented. This is a critical element for ensuring reproducibility, so we plan to address this in the future.
>
> To enhance transparency, we will release the code and dataset to the research community. Furthermore, we aim to supplement this with a tutorial or guide detailing the model architecture, hyperparameter settings, and warm-up tasks.

---

### Official Review · Reviewer_v7if · 2023-08-12

**Soundness:** 3

**Excitement:**

3: Ambivalent: It has merits (e.g., it reports state-of-the-art results, the idea is nice), but there are key weaknesses (e.g., it describes incremental work), and it can significantly benefit from another round of revision. However, I won't object to accepting it if my co-reviewers champion it.

**Paper Topic And Main Contributions:**

In short, the authors of this paper have proposed a set of warm-up tasks to prime encoder-decoder models for generating multimodal open-domain dialog. The warmup tasks are used before fine-tuning the model. They introduce the following warmup tasks - Masked Language Modeling (MLM), Masked Region Modeling (MRM), Image Swapping (ISP), and Generation Caption (GCP). The authors state that utilizing these tasks before training relatively smaller models improves performance and makes it comparable to the levels of SOTA while only utilizing the target data.

**Questions For The Authors:**

(Q)A. Does your method also improve performance on the larger version (2.7B) of Blenderbot?

(Q)B. Could you explain the large difference in the BLEU and ROUGE scores of the Dodeca model compared to your method (shown in Table 5)?

(Q)C. Regarding Table 3, it looks like the increment caused by the MRM task is incremental (or in some cases even degrading) compared to the other warm-up tasks. Could you report how much improvement the MRM task causes if no other warm-up tasks are used?

**Reasons To Accept:**

(A)A. The authors show improved performance on the Image-Chat dataset despite using a smaller model than the current SOTA on 2/3 metrics.

(A)B. The work tries to introduce a method that will be useful in improving smaller models in resource-constrained settings.

(A)C. The authors have conducted detailed ablation studies explaining the impact of each warm-up task on overall performance.

(A)D. The authors have clarified the impact of the caption generation task, which is the most significant contributor amongst the warm-up tasks.

**Reasons To Reject:**

(R)A. The authors claim that their warm-up process aligns text and images using only the target data without any pretraining. However, they utilize the BLIP-2 model for generating captions from target data which they then use in the Generation Captioning Warmup task as well as in the ablation experiment shown in Table 4. The BLIP-2 model is a large model which has been pretrained on a large amount of data (Both the Visual and Language components). They also show in Table 4 that adding captions has a larger improvement in performance (F1-score) compared to the increment created by the other three warmup tasks (MLM, MRM, and ISP). This means that the greatest improvement is caused by using captions generated by a pretrained model. Thus I think the claim that they do not use any pretraining or data sources is not entirely valid.

(R)B. The authors do not show any examples of captions generated by their model. They only report quantitative findings based on the F-1, BLEU, and ROUGE metrics. The authors should include a few examples of good and bad dialog utterances generated by their method on the image-chat dataset. It might also be a good idea to include some captions that were automatically generated for GCP.

(R)C. There is no explicit mention of what happens when their method is used with the large version of Blenderbot. Is this method only useful for smaller models? It stands to reason that their method should also improve the performance of the 2.7B version of Blenderbot. The authors should clarify whether this is the case.

**Reproducibility:**

4: Could mostly reproduce the results, but there may be some variation because of sample variance or minor variations in their interpretation of the protocol or method.

**Reviewer Confidence:**

3: Pretty sure, but there's a chance I missed something. Although I have a good feel for this area in general, I did not carefully check the paper's details, e.g., the math, experimental design, or novelty.

**Typos Grammar Style And Presentation Improvements:**

A minor grammar error in the abstract -

Moreover, our proposed approach automatically generate captions for images and incorporate them into the model’s input to improve the contextualization of visual information. -> Moreover, our proposed approach automatically generates caption*s* for images and incorporate*s* them into the model’s input to improve the contextualization of visual information.

---

> ### Author Rebuttal · Authors · 2023-08-29
>
> Thank you very much for taking the time to review our paper. I greatly appreciate your constructive comments and suggestions, which I believe have significantly improved the quality of my work. I have addressed your queries below:
> * * *
> > (R)A. The authors claim that their warm-up process aligns text and images using only the target data without any pretraining.
>
> The primary objective of this paper is to construct an effective multi-modal dialogue model utilizing solely the target domain dataset. A pivotal step in achieving this involves extracting robust representations from images for multi-modal tasks. Conventionally, this representation extraction relies on pre-trained models familiarized with the relevant domain. In line with this tradition, our study introduces the pre-trained CLIP and BLIP-2 models, effectively harnessing them for image representation and caption information extraction.
>
> The deployment of these pre-trained models is instrumental in elevating the performance of another significant innovation presented in this research: the warm-up tasks and the integrated framework. This design choice grants us the flexibility to seamlessly upgrade the model as more advanced pre-trained models emerge.
>
> Consequently, this research proposes a technique to enhance the performance of a multi-modal dialogue model using only the target data, without dependence on other pre-trained datasets or multi-task data. Incorporating pre-trained models is a key strategy to realize this ambition, and it aligns harmoniously with the overarching aim of our investigation.
> * * *
> > (R)B. The authors do not show any examples of captions generated by their model.
>
> The VLAW-MDM framework presents a new strategy for efficiently training multi-modal dialogue models. The initial goal of this research was to quantitatively evaluate and validate the performance of the algorithm. While we have verified the multi-modal learning capabilities of the model through quantitative evaluation, we recognize that these numbers alone do not provide sufficient validation for the model's interpretability and clarity.
>
> In particular, this framework integrates not only image and text information but also automatically generated caption information into the model. If the accuracy of such captions is lacking, it could negatively affect the quality of the generated dialogue and the overall user experience. Therefore, qualitative evaluation is also necessary and will be an important consideration in the next stage of research.
>
> To conduct an in-depth analysis of this issue, we plan to add a section in the paper's appendix that 'analyzes and categorizes error types associated with inaccurate caption information.' Additionally, we also intend to include a qualitative evaluation section in the paper.
> * * *
> > (Q)A. Does your method also improve performance on the larger version (2.7B) of Blenderbot?
>
> Our paper emphasizes the use of the VLAW-MDM strategy to boost the performance of small models, especially in resource-limited settings. This focus, however, does not imply the method's inapplicability to larger models. The primary objective of the VLAW-MDM strategy is to enhance performance by efficiently leveraging data. Therefore, in theory, its benefits should extend to models of any size.
>
> In the scope of our current research, we limited our experiments to smaller models due to constraints in our environment: warm-task learning and fine-tuning with BlenderBot 2.7B were not viable. Future research will delve into the strategy's effectiveness on larger models. Given the notable performance boost observed in the 400M BlenderBot versus the 110M BART model, as demonstrated in Table 3, we anticipate similar enhancements with the 2.7B version of BlenderBot. We hope that by releasing our model code, the broader research community will be better positioned to evaluate and adopt our methodology.
>
> * * *
> > (Q)B. Could you explain the large difference in the BLEU and ROUGE scores of the Dodeca model compared to your method (shown in Table 5)?
>
> The Dodeca model's performance enhancement primarily stems from an exhaustive hyper-parameter search tailored for the Image-Chat target dataset, as well as specific token generation settings. These findings are detailed in the MultiModal BlenderBot paper[1]. In particular, we found that a shorter minimum generation length boosts the BLEU-4 score, while a longer minimum generation length (50 tokens) is favorable for the ROUGE-L score.
>
> Conversely, in our research, we aligned with the MultiModal BlenderBot study[1], adopting BlenderBot's default settings and implementing our VLAW-MDM strategy (referred to as "Our approach") on top of that foundation. This means our framework enhances the model's performance through our proposed warm-up tasks, without the necessity for the parameter search that Dodeca undertook. Remarkably, we witnessed a performance surge even in a more compact BlenderBot version (400M) in comparison to the MultiModal BlenderBot (2.7B) — a claim supported by the data in Table 5.
>
> Consequently, the variances in BLEU-4 and ROUGE-L scores can be attributed largely to the differing research methodologies and settings between Dodeca and our approach. It's worth noting that these differences arise naturally from the distinct objectives and scopes characterizing each methodology.
>
> [1] Shuster, Kurt, et al. "Multi-Modal Open-Domain Dialogue." Proceedings of the 2021 Conference on Empirical Methods in Natural Language Processing. 2021.
>
> * * *
> > (Q)C. Regarding Table 3, it looks like the increment caused by the MRM task is incremental (or in some cases even degrading) compared to the other warm-up tasks. Could you report how much improvement the MRM task causes if no other warm-up tasks are used?
>
> As shown in Table 3, MRM (Masked Region Modeling) sometimes exhibits reduced efficacy when combined with other warm-up tasks. This could indicate that the MRM task might be less crucial for enhancing Image-Chat performance compared to other tasks. Nevertheless, this observation doesn't undermine the intrinsic value of the MRM task.
>
> We intend to further probe the effects of MRM in isolation, without integrating other warm-up tasks, through subsequent experiments. These will provide clearer insights into the standalone impact of MRM.

---

### Official Review · Reviewer_CkdE · 2023-08-12

**Soundness:** 3

**Excitement:**

3: Ambivalent: It has merits (e.g., it reports state-of-the-art results, the idea is nice), but there are key weaknesses (e.g., it describes incremental work), and it can significantly benefit from another round of revision. However, I won't object to accepting it if my co-reviewers champion it.

**Paper Topic And Main Contributions:**

The paper presents a new learning strategy called vision-language warm-up tasks for multimodal dialogue models. This strategy doesn't require the use of large pre training or multi-task datasets but rather relies solely on learning from target data. Moreover, the approach automatically generate captions for images and incorporate them into contextualization of visual information.

The following are the main contributions of the paper -

1. A framework for vision-language warm-up tasks in multimodal dialogue models, called VLAW-MDM, and describe the process of warming up the model using only data from the target task.

2. The paper introduces four different warm-up tasks (MLM, ISP, MRM, and GCD) and experimentally evaluated how they each affected the performance of the model.The results show that the best performance was achieved when all four warm-up tasks were utilized together.

3. The results show that the framework incorporating caption information was effective for training a multimodal open-dialog model.

4. It also evaluates warm up tasks in the absence of caption information.

**Questions For The Authors:**

Q1. Can you provide human evaluation for a small dataset with your approach?

**Reasons To Accept:**

The following are the reasons to accept the paper -

1. The paper is well-written and easy to follow.

2. It provides detailed study for the impact of each of the warm-up tasks.

3. The paper also provides a comparison for the Image-Chat with the existing model and their approach.

**Reasons To Reject:**

The following are the weakness of the paper -

1. It is missing human evaluation which is important for the evaluation.

2. The model is dependent on the quality of the generated caption which might affect the performance in real situations.


**Reproducibility:**

1: Could not reproduce the results here no matter how hard they tried.

**Reviewer Confidence:**

3: Pretty sure, but there's a chance I missed something. Although I have a good feel for this area in general, I did not carefully check the paper's details, e.g., the math, experimental design, or novelty.

---

> ### Author Rebuttal · Authors · 2023-08-29
>
> Thank you very much for taking the time to review our paper. I greatly appreciate your constructive comments and suggestions, which I believe have significantly improved the quality of my work. I have addressed your queries below:
> * * *
> > (R) 2.The model is dependent on the quality of the generated caption which might affect the performance in real situations.
>
> The VLAW-MDM framework introduces a new strategy for the efficient training of multi-modal dialogue models. Our initial objective was to quantitatively evaluate and validate the algorithm's performance. While we've confirmed the multi-modal learning capabilities of the model through quantitative evaluations, we acknowledge that these metrics alone do not provide a comprehensive validation of the model's interpretability and clarity.
>
> This framework not only integrates image and text information but also incorporates automatically generated caption data. If the captions are inaccurate, they could adversely affect the quality of the generated dialogue and the overall user experience. As a result, qualitative assessments are essential and will be a primary focus in the next research phase.
>
> We plan to include a section in the paper's appendix that 'analyzes and categorizes error types associated with inaccurate caption information.'
>
> Additionally, to understand the dependency on caption quality in our proposed model, we'll examine the dialogue generation results, differentiating between instances where captions are well-generated and those where they are not.
>
> We commit to sharing all analysis results transparently with the research community. We believe that this will spotlight areas requiring attention in future studies.
> * * *
> > (Q) 1. Can you provide human evaluation for a small dataset with your approach?
>
> We aim to introduce criteria for qualitative assessment based on frequent questions from reviewers. To this end, we are designing a framework for qualitative evaluation. Using this framework, we will carry out qualitative assessments. The evaluation criteria will focus on the naturalness of utterances, the image's reflection in the dialogue, and the accurate representation of style. We intend to score the model based on each of these criteria. We believe that this qualitative assessment will address many reviewers' concerns, and we will incorporate the results in our paper.
>
> We will transparently share the code, datasets, and experimental results for all models with the research community.

---

### Meta-Review · Area_Chair_c7iR · 2023-09-15

**Recommendation:** 3

**Metareview:**

This paper introduces a novel learning strategy termed vision-language warm-up tasks for multimodal dialogue models (VLAW-MDM). Remarkably, this strategy avoids dependence on large pretraining or multi-task datasets and instead focuses on learning directly from the target data. The authors propose four distinct warm-up tasks. Furthermore, VLAW-MDM can autonomously generate captions for images, integrating them into the model's input to better contextualize visual information. Within constraints of limited data and smaller model configurations, experimental results indicate that VLAW-MDM offers comparable, and occasionally superior, performance against existing state-of-the-art models on the ImageChat benchmark.

**Soundness Scores**: (3, 3, 3, 3)
The presented work exhibits moderate soundness. While the authors have a detailed ablation study for the four warm-up tasks, the experiments are confined to a single benchmark. Notably absent are a case study and error analysis. Additionally, the methodology section lacks a thorough exposition, potentially impeding reproducibility.

**Excitement Scores**: (3, 3, 3, 3)
The excitment towards the paper is moderate. The majority of reviewers appreciate the novel direction of relying solely on target data for pretraining and recognize the uniqueness of the techniques introduced. However, the performance improvements observed are marginal. The general applicability of the proposed techniques remains questionable due to an absence of supporting justification. This gap makes it challenging to ascertain the broader utility of the introduced methods.

In summary, this work is **moderately sound and moderately exciting** .

---

### Decision · Program_Chairs · 2023-10-07

**Decision:**

Accept-Main

**Comment:**

This paper introduces a novel learning strategy termed vision-language warm-up tasks for multimodal dialogue models (VLAW-MDM). Remarkably, this strategy avoids dependence on large pretraining or multi-task datasets and instead focuses on learning directly from the target data. The authors propose four distinct warm-up tasks. Furthermore, VLAW-MDM can autonomously generate captions for images, integrating them into the model's input to better contextualize visual information. Within constraints of limited data and smaller model configurations, experimental results indicate that VLAW-MDM offers comparable, and occasionally superior, performance against existing state-of-the-art models on the ImageChat benchmark.

**Soundness Scores**: (3, 3, 3, 3)
The presented work exhibits moderate soundness. While the authors have a detailed ablation study for the four warm-up tasks, the experiments are confined to a single benchmark. Notably absent are a case study and error analysis. Additionally, the methodology section lacks a thorough exposition, potentially impeding reproducibility.

**Excitement Scores**: (3, 3, 3, 3)
The excitment towards the paper is moderate. The majority of reviewers appreciate the novel direction of relying solely on target data for pretraining and recognize the uniqueness of the techniques introduced. However, the performance improvements observed are marginal. The general applicability of the proposed techniques remains questionable due to an absence of supporting justification. This gap makes it challenging to ascertain the broader utility of the introduced methods.

In summary, this work is **moderately sound and moderately exciting** .